# Real-World Treatment Patterns and Survival Outcomes for Patients with Non-Metastatic Non-Small-Cell Lung Cancer in Sweden: A Nationwide Registry Analysis from the I-O Optimise Initiative

**DOI:** 10.3390/cancers16091655

**Published:** 2024-04-25

**Authors:** Gudrun N. Oskarsdottir, Erik Lampa, Anders Berglund, Linda Rosengren, Maria Ulvestad, Miklos Boros, Melinda J. Daumont, Caroline Rault, Gabrielle Emanuel, Cátia Leal, Minouk J. Schoemaker, Gunnar Wagenius

**Affiliations:** 1Department of Respiratory Medicine and Allergology, Skåne University Hospital, V/O Hjärt- och Lungmedicin, 222 42 Lund, Sweden; 2Division of Oncology, Department of Clinical Sciences Lund, Lund University, Medicon Village, 22 381 Lund, Sweden; 3Epistat AB, 753 30 Uppsala, Sweden; erik.lampa@epistat.se (E.L.); anders.berglund@epistat.se (A.B.); 4Medical Department Sweden, Bristol Myers Squibb, Hemvärnsgatan 9, 171 23 Solna, Sweden; 5Medical Department Nordics, Bristol Myers Squibb, Lysaker Torg 35, 1366 Lysaker, Norway; maria.ulvestad@bms.com; 6Department of Cardiothoracic Surgery, Linköping University Hospital, 581 85 Linköping, Sweden; miklos.boros@regionostergotland.se; 7Worldwide Health Economics & Outcomes Research, Bristol Myers Squibb, 1420 Braine-L’Alleud, Belgium; melinda.daumont@gmail.com; 8Data Gnosis, 35200 Rennes, France; caroline.rault@bms.com; 9Real-World Data Analytics Markets, Bristol Myers Squibb, Uxbridge Business Park, Sanderson Road, Uxbridge UB8 1DH, UK; gabrielle.emanuel@bms.com; 10IQVIA, 2740-265 Porto Salvo, Portugal; catia.leal@iqvia.com; 11IQVIA, 1101 CT Amsterdam, The Netherlands; minouk.schoemaker@iqvia.com; 12Department of Oncology-Pathology, Karolinska Institute, 141 86 Stockholm, Sweden; gunnar.wagenius@regionstockholm.se; 13Thoracic Oncology Centre, The Cancer Theme, Karolinska University Hospital, 171 76 Stockholm, Sweden

**Keywords:** I-O Optimise, non-small-cell lung cancer (NSCLC), overall survival, time to next treatment or death, real-world evidence, registry, Sweden

## Abstract

**Simple Summary:**

Non-small-cell lung cancer (NSCLC) is the leading cause of cancer-related death worldwide, with almost half of all patients diagnosed with non-metastatic disease (stage IA–IIIC). The treatment options for patients with NSCLC are evolving rapidly, and survival outcomes have improved since the introduction of immunotherapies and targeted treatments in the non-metastatic setting. In this study, we explored treatment patterns and outcomes for patients with non-metastatic NSCLC in Sweden prior to the availability of these treatments. Patient outcomes were comparable with those reported in other real-world studies; however, the prognosis for patients with NSCLC, particularly at higher disease stages, remained poor. These results provide a baseline upon which to evaluate the effectiveness of immunotherapies and targeted treatments as they are introduced into routine clinical practice, including for patients in the non-metastatic setting.

**Abstract:**

Non-small-cell lung cancer (NSCLC) is the leading cause of cancer-related death worldwide, with ~40–50% of patients diagnosed with non-metastatic disease (stages IA–IIIC). The treatment landscape is evolving rapidly as immunotherapies and targeted therapy are introduced in the non-metastatic setting, creating a need to assess patient outcomes prior to their introduction. This real-world study using Swedish National Lung Cancer Registry data examined outcomes (overall survival (OS) and time to next treatment or death (TTNTD)) and treatment patterns for adults diagnosed with non-metastatic NSCLC. Baseline characteristics and OS from diagnosis were described for all patients; OS, treatment patterns, and TTNTD from treatment start were described for the treatment subgroup (patients diagnosed from 2014 onwards), stratified by disease stage and initial treatment. OS and TTNTD were described using the Kaplan–Meier estimator. The overall population (2008–2019) included 17,433 patients; the treatment subgroup included 5147 patients. Median OS (interquartile range) overall ranged from 83.3 (31.6–165.3) months (stage I patients) to 10.4 (4.3–24.2) months (stage IIIB patients). Among the treatment subgroup, median OS and TTNTD were longest among patients receiving surgery versus other anticancer treatments. These findings provide a baseline upon which to evaluate the epidemiology of non-metastatic NSCLC as newer treatments are introduced.

## 1. Introduction

Lung cancer is the leading cause of cancer-related mortality, accounting for around 18% of cancer deaths globally; of all lung cancer cases, approximately 85% are attributed to non-small-cell lung cancer (NSCLC) [1,2]. In 2022, there were 4466 new lung cancer cases recorded in Sweden, with lung cancer accounting for around 7% of all diagnosed cancers and approximately 16% of cancer-related deaths [3,4]. Worldwide, approximately 20–30% of patients with NSCLC are diagnosed with early-stage disease (stage I/II), around 30% have locally advanced disease (stage III), and 50% have metastatic disease (stage IV) [5]. Registry data covering the period between 2020 and 2022 in Sweden reported that at diagnosis, 25.8% of patients with NSCLC had stage I, 6.6% had stage II, 18.0% had stage III, and 48.7% had stage IV disease (1.0% had unknown disease stage) [4]. Historically, the prognosis for patients with NSCLC has been poor, with global 5-year overall survival (OS) rates of 68.0–92.0% for patients with stage I, 53.0–60.0% for patients with stage II, and 13.0–36.0% for patients with stage III disease, decreasing to <10% for those with metastatic disease (stage IV) [6]. Furthermore, over a 5-year follow-up period, for those patients with NSCLC who are eligible for surgery, more than 30% experience disease recurrence, even after complete resection (almost half of whom received adjuvant chemotherapy) [7]. Among patients with inoperable stage III NSCLC receiving chemoradiotherapy, the prognosis is worse, with only 15% of patients alive after 5 years [8].

Over the past decade, the treatment guidelines for non-metastatic NSCLC have recommended surgery for patients with resectable tumors, with (neo)adjuvant chemotherapy advised for specific patient populations [9,10]. For patients with unresectable tumors, radiotherapy (RT) has been the recommended standard of care for those with stage I NSCLC, and concurrent or sequential chemoradiotherapy has been the recommended standard of care for those with stage III NSCLC [9]. More recently, immunotherapy (IO)-based and targeted therapies in the (neo)adjuvant setting have been shown to improve patient outcomes in several clinical trials [11,12,13]. Approvals by the European Medicines Agency and US Food and Drug Administration have resulted in updated guidelines recommending platinum-based chemotherapy in combination with an immune checkpoint inhibitor (ICI) as neoadjuvant treatment, and ICIs as adjuvant treatment for patients with resectable NSCLC [14,15,16]. An ICI is also now approved after concurrent chemoradiation for patients with locally advanced, unresectable disease [14,16]. The use of IO for neoadjuvant treatment has not yet been adopted in Sweden; however, it is anticipated that this will be incorporated into the next edition of the care program, which is due in 2024. Furthermore, the feasibility of large-scale screening programs for individuals with a high risk of developing lung cancer is currently being explored in Sweden, with the aim of introducing routine screening at a national level [17]. As early detection methods such as screening are more widely adopted, the number of patients being diagnosed early will inevitably increase, highlighting the importance of treatments that can target the initial stages of disease [18].

In consideration of the rapidly expanding treatment landscape for non-metastatic NSCLC, there is a need to evaluate the effectiveness of these new treatments outside of clinical trial settings; an important part of this evaluation is the establishment of a real-world baseline against which future changes in patient management and clinical outcomes can be compared [19]. I-O Optimise is an ongoing, multicountry, collaborative research initiative leveraging existing real-world cancer databases to provide insights into the management of thoracic malignancies in clinical practice [19]. As part of the I-O Optimise initiative, the objective of this study was to describe patient characteristics, treatment patterns, and survival outcomes among patients diagnosed with non-metastatic NSCLC using real-world data from population-based national health registries in Sweden prior to the availability of newer treatment options.

## 2. Methods

### 2.1. Inclusion Criteria and Study Design

This was a retrospective, nationwide real-world study with a cohort design using existing data collected from the National Lung Cancer Registry (NLCR) in Sweden. Included patients were aged ≥18 years with incident stage I–IIIC NSCLC at diagnosis and an International Classification of Diseases—10th revision (ICD-10) code of C34.0, C34.1, C34.2, C34.3, or C34.8 (C34.9 was not captured in the registry data) between 1 January 2008 and 31 December 2019. Patients with missing data on age at diagnosis or sex were excluded. Due to changes in how data were collected by the NLCR in 2014, treatment exposure data were only available from 2014; therefore, patients were categorized into two groups based on their date of diagnosis. The overall population group (patients diagnosed between 1 January 2008 and 31 December 2019) was analyzed for patient demographics and survival outcomes. The treatment subgroup (patients diagnosed between 1 January 2014 and 31 December 2019) was analyzed for treatment patterns, OS, and time to next treatment or death (TTNTD). Patients were followed from their initial NSCLC diagnosis (inclusion date) until the earliest of death, exit from the data source, or censoring at the end of the study’s follow-up period, whichever occurred first. Stratification analyses included disease stage at diagnosis, initial treatment type, histology, and date of diagnosis.

### 2.2. Variables

Demographic and clinical characteristics (age, sex, smoking status, and Eastern Cooperative Oncology Group (ECOG) performance status) were collected for all patients at diagnosis. Patients were stratified according to stage of disease at initial diagnosis: stage I (including IA and IB), II (including IIA and IIB), IIIA, IIIB, and IIIC. Histology at diagnosis (defined according to the International Classification of Diseases for Oncology—2nd edition (ICD-O-2) morphology codes) was categorized into the following groups: non-squamous NSCLC (adenocarcinoma and large cell carcinoma), squamous cell NSCLC, and NSCLC not otherwise specified, and was recorded for all patients. Tumor, node, and metastasis (TNM) staging followed the American Joint Committee on Cancer (AJCC)/Union for International Cancer Control (UICC) staging manual, 6th, 7th, and 8th editions according to the time of patient inclusion in the NLCR. As this study was performed over a 12-year period, TNM staging was categorized according to the recommended edition at the time the patient was diagnosed. Stages IA1, IA2, IA3, and IIIC were included for the first time in the AJCC/UICC 8th edition.

The type of initial treatment received was categorized according to whether any surgery, any RT, or any systemic anticancer therapy (SACT) was received. Initial treatments received were defined according to a treatment algorithm (Table A1) into the following groups: surgery alone, surgery with adjuvant SACT and/or RT, surgery with perioperative SACT and/or neoadjuvant SACT with or without RT, RT alone, palliative SACT with RT, sequential or concurrent chemoradiation, other curative SACT with RT, other SACT with RT, SACT alone, no treatment received, and otherwise unclassified treatment. The time-to-event endpoints analyzed included OS from diagnosis, OS from start of initial treatment, and TTNTD.

### 2.3. Statistical Analysis

Patient demographic and clinical characteristics were described using summary statistics. Primary data masking was performed if patient counts for individual categories were greater than zero but fewer than five. OS and TTNTD, stratified by disease stage and initial treatment received, were described using Kaplan–Meier estimator. Data are not shown for any treatment category where all data are masked and/or the number of patients recorded as receiving the treatment at each stage was fewer than 20.

### 2.4. Ethics

This study was conducted in accordance with the International Society for Pharmacoepidemiology (ISPE) Guidelines for Good Pharmacoepidemiology Practices (GPP) and was approved by the Swedish Ethical Review Authority (Etikprövningsmyndigheten). Informed patient consent was not required due to the retrospective nature of the study.

## 3. Results

### 3.1. Patient Population

In the overall treatment group, 17,433 patients diagnosed with stage I–IIIC NSCLC were included; the largest proportion of patients were diagnosed with stage I disease (41.0%; n = 7153). There were 2675 patients (15.3%) diagnosed with stage II, 3679 (21.1%) with stage IIIA, 3727 (21.4%) with stage IIIB, and 199 (1.1%) with stage IIIC disease (Table 1). The median age (interquartile range (IQR)) of patients was similar across all disease stages (71.0–72.0 (64.0–77.0) years), and most patients were smokers or former smokers (87.2–91.5%). Across all stages, a majority of patients had an ECOG performance status ≤ 1 (stage I, 88.0%; stage II, 77.6%; stage IIIA, 73.2%; stage IIIB, 61.8%; stage IIIC, 63.3%) (Table 1). Adenocarcinoma was the most frequently recorded histology in all groups (stage I, 71.8%; stage II, 53.8%; stage IIIA, 50.0%; stage IIIB, 50.4%; stage IIIC, 45.7%) (Table 1). In the treatment subgroup, a total of 5147 patients were included (diagnosed from 2014 onward), of whom 2259 (43.9%) had stage I, 819 (15.9%) had stage II, 1051 (20.4%) had stage IIIA, 879 (17.1%) had stage IIIB, and 139 (2.7%) had stage IIIC disease (Table 2).

### 3.2. Treatment Patterns

In the treatment subgroup, surgery was the most frequently observed initial treatment in patients with stage I and stage II disease (n = 1578 (69.9%) and n = 460 (56.2%), respectively). In patients with stage I disease, surgery alone was more frequently recorded than surgery with adjuvant SACT ± RT (n = 1269 (56.2%) vs. n = 308 (13.6%)), and fewer than five patients received surgery with perioperative SACT or neoadjuvant SACT ± RT. RT alone was recorded for 488 patients (21.6%) and no treatment was recorded within 180 days of diagnosis for 114 patients (5.0%) (Table 2, Figure 1).

For patients with stage II disease, surgery with adjuvant SACT ± RT was more frequently recorded than surgery alone (n = 272 (33.2%) vs. n = 179 (21.9%)), surgery with perioperative SACT or neoadjuvant SACT ± RT was recorded for nine patients (1.1%), and RT alone was recorded for 127 patients (15.5%). Concurrent chemoradiation was received by 48 patients (5.9%) and sequential chemoradiation was received by nine patients (1.1%). No treatment within 180 days of diagnosis was recorded for 87 patients (10.6%) (Table 2, Figure 1).

Among patients with stage III disease, 184 (17.5%) with stage IIIA, 16 (1.8%) with stage IIIB, and zero with stage IIIC disease received any surgery as initial treatment. Any SACT was the most frequently recorded initial treatment in all subgroups (stage IIIA, n = 396 (37.7%); stage IIIB, n = 399 (45.4%); stage IIIC, n = 65 (46.8%)) (Table 2).

Of the patients with stage IIIA NSCLC, surgery alone was received by 67 patients (6.4%), surgery with adjuvant SACT ± RT was received by 100 patients (9.5%), and surgery with perioperative SACT or neoadjuvant SACT ± RT was received by 17 patients (1.6%). RT alone was received by 112 (10.7%) patients with stage IIIA. Concurrent and sequential chemoradiation were recorded for 239 (22.7%) and 44 (4.2%) patients with stage IIIA disease, respectively, and SACT alone was received by 185 patients (17.6%); palliative SACT with RT was recorded for 52 patients (4.9%), and 17.3% of patients (n = 182) in this group received no treatment within 180 days of their diagnosis (Table 2, Figure 1).

Among patients with stage IIIB NSCLC, fewer than 5 received surgery alone or surgery with perioperative SACT or neoadjuvant SACT ± RT. Surgery with adjuvant SACT ± RT was received by 11 patients (1.3%) and RT alone was received by 96 (10.9%). Concurrent and sequential chemoradiation were recorded for 154 (17.5%) and 45 (5.1%) patients, respectively. SACT alone was recorded for 277 patients (31.5%); palliative SACT with RT was recorded for 85 patients (9.7%). No treatment within 180 days of diagnosis was recorded for 181 patients (20.6%).

No patients with stage IIIC disease had surgery as initial treatment and 8 patients (5.8%) received RT alone. Concurrent chemoradiation was received by 14 patients (10.1%) and fewer than 5 patients received sequential chemoradiation. SACT alone was recorded for 42 patients (30.2%), and 49 patients (35.3%) were recorded as having received no treatment within 180 days of their diagnosis (Table 2, Figure 1).

### 3.3. Overall Survival

In the overall patient population (including all patients who may or may not have received treatment during the study period), patients with stage I disease had a median OS (IQR) from diagnosis date of 83.3 (31.6–165.3) months and 1-, 3-, and 5-year survival probabilities (95% confidence interval (CI)) of 0.90 (0.89–0.91), 0.73 (0.72–0.74), and 0.60 (0.59–0.61), respectively. Among patients with stage II disease, median OS (IQR) was 31.9 (11.5–94.3) months with 1-, 3-, and 5-year survival probabilities (95% CI) of 0.74 (0.72–0.76), 0.47 (0.45–0.49), and 0.35 (0.33–0.37), respectively. Stage IIIA patients had a median OS (IQR) of 16.7 (7.3–43.9) months with 1-, 3-, and 5-year survival probabilities (95% CI) of 0.61 (0.60–0.63), 0.30 (0.28–0.31), and 0.20 (0.18–0.21), respectively. Stage IIIB patients had a median OS (IQR) of 10.4 (4.3–24.2) months, with 1-, 3-, and 5-year survival probabilities (95% CI) of 0.45 (0.44–0.47), 0.17 (0.16–0.18), and 0.11 (0.10–0.12). Stage IIIC patients had a median OS (IQR) of 11.1 (5.0–27.4) months with 1-, 3-, and 5-year survival probabilities (95% CI) of 0.48 (0.42–0.56), 0.18 (0.13–0.24), and 0.10 (0.05–0.20) (Figure 2).

Among the treatment subgroup, median OS (IQR) from treatment start date was not reached (NR) in patients with stage I disease (39.5 months–NR; 1-, 3-, and 5-year survival probabilities (95% CI) of 0.92 (0.91–0.93), 0.77 (0.75–0.79 and 0.65 (0.63–0.67)). In patients with stage II disease, median OS (IQR) was 42.0 months (15.1 months–NR; 1-, 3-, and 5-year survival probabilities (95% CI) of 0.80 (0.77–0.83), 0.54 (0.51–0.58), and 0.40 (0.36–0.44)). Among patients with stage IIIA disease, median OS (IQR) was 27.9 (10.5–66.8) months (1-, 3-, and 5-year survival probabilities (95% CI) of 0.72 (0.69–0.75), 0.44 (0.41–0.47), and 0.28 (0.25–0.32)). In patients with stage IIIB disease, median OS was 16.2 (6.8–42.8) months (1-, 3-, and 5-year survival probabilities (95% CI) of 0.58 (0.55–0.62), 0.29 (0.26–0.33), and 0.19 (0.16–0.22)) and in patients with stage IIIC disease, median OS was 15.3 (6.7–30.8) months (1-, 3-, and 5-year survival probabilities (95% CI) of 0.60 (0.51–0.71), 0.24 (0.17–0.35), and 0.11 (0.04–0.30)) (Figure 3). Regardless of disease stage, median OS (IQR) from treatment start date was longest for patients who received surgery alone (NR (50.9 months–NR)), surgery with adjuvant SACT ± RT (83.1 (31.1 months–NR)), or surgery with neoadjuvant SACT ± RT (NR (40.7 months–NR)) versus patients in all other treatment categories (Figure 4).

Among patients with stage I–IIIA disease in the treatment subgroup, median OS (IQR) from treatment start date was longest among those who received surgery alone or surgery with adjuvant SACT ± RT (stage I, NR (60.3 months–NR) and NR (55.8 months–NR), respectively; stage II, 73.5 (19.3–NR) months and 70.2 (25.4–NR) months, respectively; stage IIIA, 59.1 (25.0–NR) months and 52.3 (20.6–NR) months, respectively).

Among patients with stage I and II disease, median OS (IQR) in those receiving RT alone was 41.2 (16.7–79.4) and 19.4 (7.3–36.9) months, respectively. Among patients with stage IIIA and IIIB disease, median OS (IQR) with RT alone was 13.9 (3.2–37.0) and 8.9 (3.6–22.0) months, respectively. Median OS (IQR) in patients receiving SACT alone was 46.8 (12.4–79.2) months in those with stage I NSCLC, 20.0 (8.6–36.4) months in stage II disease, 16.0 (7.3–39.8) months in stage IIIA disease, 13.5 (6.5–32.5) months in stage IIIB disease, and 17.4 (7.7–30.8) with stage IIIC disease (Figure A1a–e).

Among patients receiving concurrent chemoradiation, patients had a median OS (IQR) from treatment start date of 42.0 (17.2–78.8) months (stage II), 42.3 (16.6–NR) months (stage IIIA), and 40.7 (15.2–95.3) months (stage IIIB) (Figure A1b–d). Patients who received sequential chemoradiation had a median OS (IQR) of 27.4 (11.9–54.2) months (stage IIIA) and 25.4 (12.8–NR) months (stage IIIB).

### 3.4. Time to Next Treatment or Death

Among the treatment subgroup, median TTNTD (IQR) from initial treatment was 93.4 (34.8–NR) months for patients with stage I, 34.5 (12.1–NR) months for patients with stage II, 20.6 (8.0–59.7) months for patients with stage IIIA, 12.1 (5.7–32.6) months for patients with stage IIIB, and 12.5 (4.7–29.5) months for patients with stage IIIC NSCLC (Figure 5). Across all disease stages, median TTNTD (IQR) was the longest among patients receiving surgery alone (NR, 48.2 months–NR), surgery with adjuvant SACT ± RT (83.1 (20.7–NR) months), and surgery with neoadjuvant/perioperative SACT ± RT (61.4 (15.8–NR) months) (Figure 6). Median TTNTD (IQR) was the longest among patients with stage I and II NSCLC who received surgery or surgery with adjuvant SACT ± RT (stage I, NR (56.0 months–NR) and NR (41.8 months–NR), respectively; stage II, 65.7 (17.1–NR) months and 60.5 (17.5–NR) months, respectively). In both groups (stages I and II), TTNTD (IQR) was the shortest among patients who received SACT alone (stage I, 22.6 (8.6–65.4) months; stage II, 12.1 (4.2–30.3) months) (Figure A2a,b). Among patients with stage IIIA disease, median TTNTD (IQR) was longer among patients who received surgery alone (55.0 (25.0–NR) months) or concurrent chemoradiation (31.1 (12.4–83.2) months) and the shortest among patients who received palliative SACT with RT (9.7 (5.4–15.4) months) (Figure A2c). In patients with stage IIIB NSCLC, median TTNTD (IQR) was the longest in those who received concurrent chemoradiation (27.2 (11.2–NR) months) and the shortest in those who received palliative SACT with RT (8.1 (5.4–14.1) months) (Figure A2d). In patients with stage IIIC NSCLC, median TTNTD (IQR) was 16.0 (6.4–30.2) months in patients receiving SACT alone (Figure A2e).

## 4. Discussion

Using data collected from the NLCR in Sweden between 2008 and 2019, this nationwide study provides insight into the real-world outcomes for patients with non-metastatic NSCLC during this time. These insights provide a baseline upon which newer treatment options such as ICIs and tyrosine kinase inhibitors (TKIs) can be evaluated and provide an overview of patient outcomes prior to the introduction of routine lung cancer screening in Sweden. The characteristics of the patient population were consistent with the profile of patients with NSCLC in previously published real-world studies, with a median age of 70 years at diagnosis and a higher proportion of patients having adenocarcinoma compared to other subtypes [20,21,22].

The distribution of patients across initial treatment type by disease stage reported here is similar to that reported in a recent analysis of a large dataset in the US (based on Surveillance, Epidemiology, and End Results and National Program of Cancer Registries data) during the same period [20]. In this and the US study, surgery was the most common initial treatment at earlier stages of the disease (stages I and II), while SACT alone or in combination with RT was the most common initial treatment at later stages (stages IIIA and IIIB), which is consistent with guideline recommendations at the time of the study [9,14,23]. Results from the same US study [20] also revealed that the proportion of patients who did not receive treatment increased with disease stage, consistent with the results presented here. The widespread use of surgery and RT as initial treatment for stage I patients in this study was similar to observations from real-world Canadian and UK studies covering the same time period [24,25]. The proportion of patients with stage IIIA and IIIB NSCLC receiving any surgery as initial treatment (17.5% and 1.8%, respectively) was lower in this study compared with results from the multicountry KINDLE study including patients across Asia, the Middle East, Africa, and Latin America, in which 21.4% received surgery. However, the proportion of patients receiving no treatment in the KINDLE study (8.8%) was lower than shown here (17.3–20.6%), which may reflect differences in how patients in poorer health are treated [22]. Surgical resection remains controversial in patients with stage IIIB NSCLC, with many patients considered to be inoperable [5]; in this study, 16 patients (1.8%) with stage IIIB NSCLC received any surgery as part of their initial treatment. For patients with unresectable locally advanced stage IIIA or IIIB NSCLC, concurrent chemoradiotherapy is the recommended standard of care [9]. In this study, the use of concurrent chemoradiation in stage IIIA and IIIB patients was lower compared with a similar real-world population in Spain (17.5–22.7% versus 30.2–37.0%, respectively) [26]. This may reflect differences in the patient populations included in each of the studies and how data are recorded in the respective registries (e.g., whether all patients with NSCLC are included as in the NLCR or whether only patients who are being considered for treatment are included).

As expected, OS estimates decreased with increasing stage, highlighting the unmet need for improved therapeutic approaches, particularly in patients with stage II or III NSCLC. Across all disease stages, OS outcomes were longer for patients who received surgery, either alone or with (neo)adjuvant SACT ± RT; however, this is likely to reflect the larger number of patients at earlier stages of disease receiving treatment, as demonstrated in the treatment patterns by stage highlighted here. Among patients with later-stage disease, median OS from diagnosis was lower than observed in a recent study from Spain covering a similar period (stage IIIA, 37 months and stage IIIB, 28 months vs. 16.7 months and 10.4 months, respectively, presented here). However, this may also reflect the inclusion of all patients in Sweden with a diagnosis of NSCLC rather than only patients who are being considered for treatment [26]. For patients with stage IIIB disease, the results presented here were consistent with real-world findings from Portugal (10.4 months vs. 11.4 months in Portugal) [27]. OS outcomes by treatment type for patients with stage IIIA and stage IIIB NSCLC were consistent with results from the Spanish study, with broadly similar OS estimates seen for each treatment type; however, the Spanish study included patients who had received surgery with neoadjuvant SACT, which at the time of the study was not recommended for treatment in Sweden [26].

During the course of this study, the TNM staging system evolved from the 6th edition (used between 2008 and 2009) to the 7th edition (used between 2010 and 2017), and finally to the 8th edition (introduced in 2018), which inevitably affected how patients were staged during the study period [28,29]. There is evidence in other studies of upstaging and downstaging of patients with stage I and II disease between the 7th and 8th editions [30,31,32]. Furthermore, the recategorization of stage IIIB patients with pleural fluid to stage IV will have reduced the number of stage IIIB patients with a worse prognosis in this category, resulting in an apparently improved survivability for patients with stage IIIB disease [33]. In addition, stage IIIC was a new category introduced in the 8th edition, which explains the low number of patients with stage IIIC disease reported here. It is therefore possible that these changes may have affected the recording of outcomes for patients in these strata; however, as this study did not include a temporal analysis of stage at diagnosis, further analysis of whether this may have affected the results is not possible.

As expected, TTNTD was longer in patients with stage I disease, and decreased with increasing disease stage. Across the entire treated population, TTNTD was longest among patients who received surgery alone or surgery with (neo)adjuvant SACT ± RT and shortest among patients who received palliative SACT with RT or SACT alone. OS and TTNTD were similar across most treatment types; however, TTNTD was much shorter than OS for patients receiving surgery with adjuvant SACT ± RT, concurrent chemoradiation, or sequential chemoradiation. These similarities between OS and TTNTD suggest that where beneficial (e.g., when OS data are immature), TTNTD could be used as a proxy for OS [34]. This was recently demonstrated in a study focused on ICI use in patients with advanced melanoma where TTNTD was used as a surrogate endpoint. In that study, the authors found that TTNTD is particularly useful in the context of regulatory and healthcare decision making around ICIs, as it reflects the result of a therapeutic medical decision as well as taking into account the benefit the patient receives through the prolonged treatment effect [35]. The results presented here for TTNTD do not include details of treatment discontinuations or subsequent treatments; therefore, a direct comparison between the two measures should be approached with caution. However, they do provide a picture of the treatment-free interval between treatment initiation and subsequent line of treatment or death (whichever occurred first).

The treatment landscape for NSCLC is rapidly evolving, and treatment recommendations are increasingly being driven by molecular testing. The identification of bio-marker-defined NSCLC patient subgroups and associated targeted therapies have improved clinical outcomes when compared with traditional chemotherapy in patients with NSCLC [36]. Immunotherapies such as ICIs have become a cornerstone of treatment for patients with metastatic NSCLC [37]; however, the landscape for patients with non-metastatic NSCLC is now also changing. Recent approvals in the non-metastatic setting include the use of (neo)adjuvant ICIs with chemotherapy for patients with resectable NSCLC with a high risk of recurrence and programmed death ligand 1 (PD-L1) expression ≥ 1%, and the use of adjuvant ICIs with chemotherapy in patients with stage II–IIIA NSCLC and a high risk of recurrence plus PD-L1 expression ≥ 50% is a further example of how the treatment landscape is evolving with great potential for improved outcomes in patients [38,39,40]. More time is required, however, to assess the real-world impact of these newer treatments in patients with NSCLC. Establishing a baseline for treatment patterns and patient outcomes using real-world evidence can provide both valuable insight into the factors that influence clinical care pathways and accurate monitoring of these patients as newer therapies become more widely adopted in clinical practice.

A strength of this study is that the data are derived from the Swedish NLCR, which has almost complete coverage of the Swedish population (~97% of all lung cancer cases). The data available from this source provide detailed coverage of the baseline characteristics of patients for all hospitals in Sweden; however, the coverage of treatment data with available follow-up information is around 62% of the patient population, and the available data do not include details of patient ethnicity, body mass index, or socioeconomic status. Consistent with many real-world studies, a limitation of this study was the potential for variation in how data are collected in routine clinical practice, presenting the possibility of incomplete or misclassified data. Many of the analyses performed in this study were also based on groups with limited sample sizes, making it challenging to draw conclusions for these groups. The changes in TNM staging that occurred during the course of this study limited the ability to perform a direct comparison of patient outcomes by stage; some analyses were also restricted by sample sizes due to the introduction of new categories (e.g., stage IIIC was introduced in the 8th TNM staging edition), making it challenging to interpret OS data over a longer period of time. In addition, neoadjuvant and perioperative treatments were rarely recorded as these treatments were not used in Sweden at the time of this study; therefore, conclusions around their use in real-world practice could not be established. Finally, for most analyses, OS and TTNTD were calculated using the treatment start date as the index date to mitigate immortal time bias in the results; however, as some treatments were inevitably started later (e.g., for a patient to be classified as receiving surgery plus adjuvant SACT, the patient must have survived from diagnosis to surgery and from surgery to adjuvant SACT), any bias cannot be eliminated entirely, and no statistical techniques were included to mitigate this potential issue.

## 5. Conclusions

The results from this large, population-based, real-world cohort study provide insights into treatment patterns and outcomes for patients with non-metastatic NSCLC in Sweden in the period prior to the availability of newer therapies. Patient outcomes were comparable with those reported in other real-world studies; however, the prognosis for patients with NSCLC, particularly at higher disease stages, remained poor. These real-world data provide a baseline upon which to evaluate the impact of newer treatments such as ICIs and TKIs as they are incorporated into routine clinical practice.

## Figures and Tables

**Figure 1 cancers-16-01655-f001:**
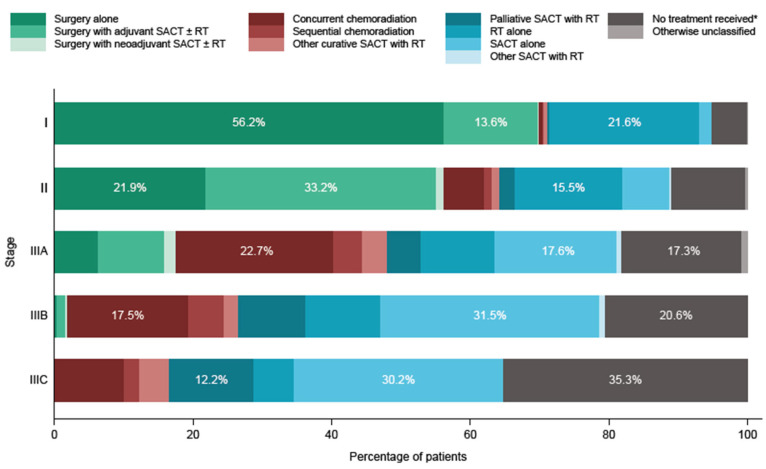
Initial treatment received by patients with stage I–IIIC NSCLC (%) in the treatment subgroup (n = 5147). * No treatment received within 180 days of diagnosis. Percentages are shown for the three most commonly received initial treatments at each disease stage. RT: radiotherapy; SACT: systemic anticancer therapy.

**Figure 2 cancers-16-01655-f002:**
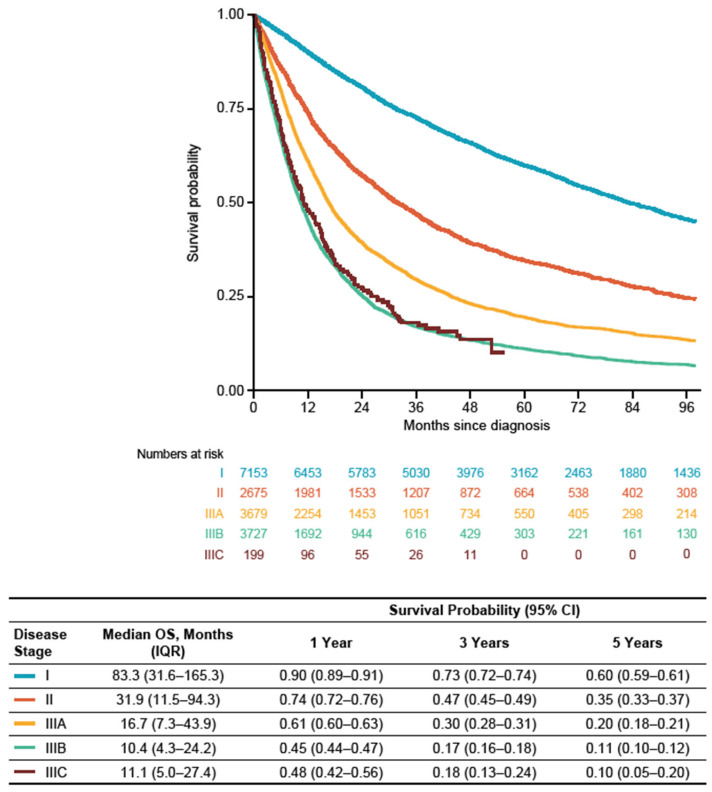
OS from date of diagnosis by stage at diagnosis among patients in the overall population (patients diagnosed between 1 January 2008 and 31 December 2019). CI: confidence interval; IQR: interquartile range; OS: overall survival.

**Figure 3 cancers-16-01655-f003:**
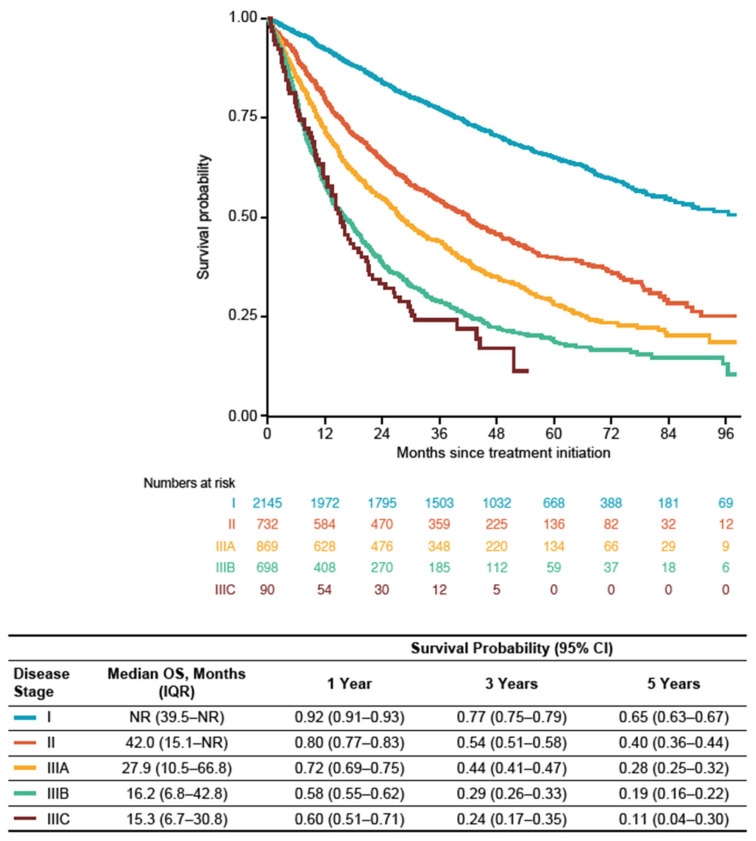
OS from treatment start date by stage at diagnosis in the treatment subgroup (patients diagnosed between 1 January 2014 and 31 December 2019). CI: confidence interval; IQR: interquartile range; NR: not reached; OS: overall survival.

**Figure 4 cancers-16-01655-f004:**
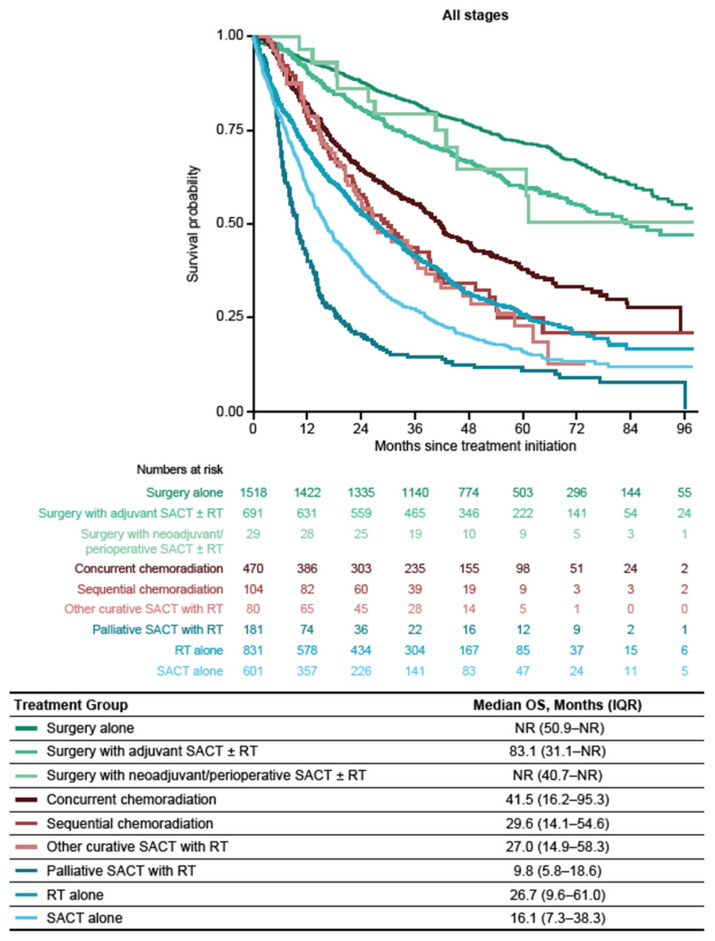
OS from treatment start date by treatment type in the treatment subgroup (patients diagnosed between 1 January 2014 and 31 December 2019). IQR: interquartile range; NR: not reached; OS: overall survival; RT: radiotherapy; SACT: systemic anticancer therapy.

**Figure 5 cancers-16-01655-f005:**
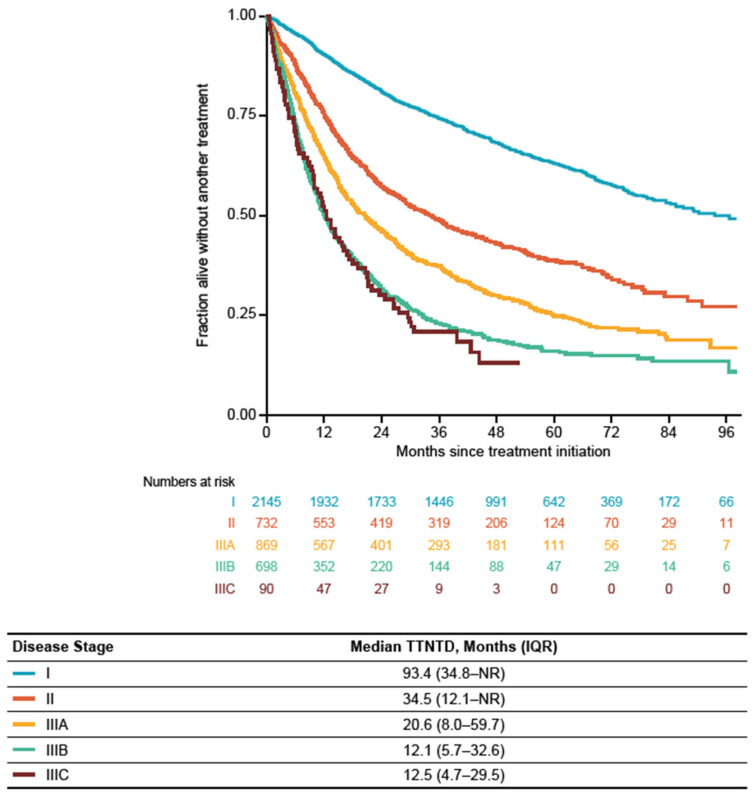
Time to next treatment or death from initial treatment by stage at diagnosis in the treatment subgroup (patients diagnosed between 1 January 2014 and 31 December 2019). IQR: interquartile range; NR: not reached; TTNTD: time to next treatment or death.

**Figure 6 cancers-16-01655-f006:**
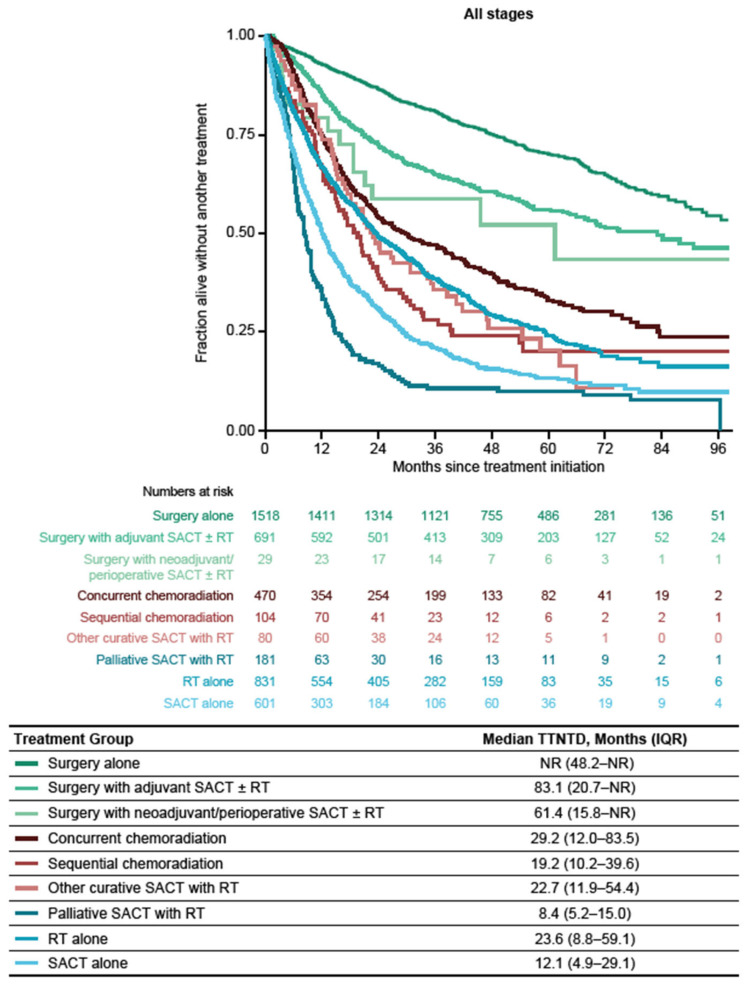
Time to next treatment or death from initial treatment by treatment type among patients in the treatment subgroup (patients diagnosed between 1 January 2014 and 31 December 2019). IQR: interquartile range; NR: not reached; RT: radiotherapy; SACT: systemic anticancer therapy; TTNTD: time to next treatment or death.

**Table 1 cancers-16-01655-t001:** Baseline characteristics for the overall patient population.

Overall Patient Population N = 17,433	Stage I	Stage II	Stage IIIA	Stage IIIB	Stage IIIC
	(n = 7153)	(n = 2675)	(n = 3679)	(n = 3727)	(n = 199)
Age at diagnosis, years					
Median (IQR)	71.0 (65.0–76.0)	71.0 (65.0–77.0)	71.0 (65.0–77.0)	71.0 (64.0–77.0)	72.0 (65.5–77.0)
Female, n (%)	3988 (55.8)	1257 (47.0)	1833 (49.8)	1725 (46.3)	96 (48.2)
Smoking status, n (%)					
Smoker	2715 (38.0)	1106 (41.3)	1544 (42.0)	1607 (43.1)	72 (36.2)
Former smoker	3523 (49.3)	1341 (50.1)	1805 (49.1)	1786 (47.9)	106 (53.3)
Never smoker	885 (12.4)	219 (8.2)	313 (8.5)	307 (8.2)	21 (10.6)
NA	30 (0.4)	9 (0.3)	17 (0.5)	27 (0.7)	0 (0.0)
ECOG performance status at diagnosis, n (%)					
0	3491 (48.8)	976 (36.5)	1137 (30.9)	790 (21.2)	51 (25.6)
1	2805 (39.2)	1099 (41.1)	1556 (42.3)	1513 (40.6)	75 (37.7)
2	654 (9.1)	393 (14.7)	604 (16.4)	836 (22.4)	43 (21.6)
3	134 (1.9)	141 (5.3)	266 (7.2)	386 (10.4)	23 (11.6)
4	21 (0.3)	29 (1.1)	52 (1.4)	104 (2.8)	<5
NA	48 (0.7)	37 (1.4)	64 (1.7)	98 (2.6)	6 (3.0)
Histology, n (%)					
Adenocarcinoma	5134 (71.8)	1440 (53.8)	1841 (50.0)	1877 (50.4)	91 (45.7)
Large cell carcinoma	344 (4.8)	189 (7.1)	387 (10.5)	441 (11.8)	19 (9.5)
Squamous cell NSCLC	1513 (21.2)	943 (35.3)	1321 (35.9)	1278 (34.3)	83 (41.7)
NSCLC, not otherwise specified	162 (2.3)	103 (3.9)	130 (3.5)	131 (3.5)	6 (3.0)

Abbreviations: ECOG: Eastern Cooperative Oncology Group; IQR: interquartile range; NA: not available; NSCLC: non-small-cell lung cancer.

**Table 2 cancers-16-01655-t002:** Initial treatment received for the treatment subgroup.

Treatment Subgroup	Stage I	Stage II	Stage IIIA	Stage IIIB	Stage IIIC
N = 5147	(n = 2259)	(n = 819)	(n = 1051)	(n = 879)	(n = 139)
Treatment type, n (%)					
Any surgery as initial treatment	1578 (69.9)	460 (56.2)	184 (17.5)	16 (1.8)	0 (0.0)
Any RT as initial treatment	525 (23.2)	196 (23.9)	245 (23.3)	211 (24.0)	31 (22.3)
Any SACT as initial treatment	365 (16.2)	359 (43.8)	396 (37.7)	399 (45.4)	65 (46.8)
Initial treatment received, n (%)					
Surgery alone	1269 (56.2)	179 (21.9)	67 (6.4)	<5	0 (0.0)
Surgery with adjuvant SACT ± RT	308 (13.6)	272 (33.2)	100 (9.5)	11 (1.3)	0 (0.0)
Surgery with perioperative SACT or neoadjuvant SACT ± RT	<5	9 (1.1)	17 (1.6)	<5	0 (0.0)
RT alone	488 (21.6)	127 (15.5)	112 (10.7)	96 (10.9)	8 (5.8)
Palliative SACT with RT	9 (0.4)	18 (2.2)	52 (4.9)	85 (9.7)	17 (12.2)
Sequential chemoradiation	<5	9 (1.1)	44 (4.2)	45 (5.1)	<5
Concurrent chemoradiation	15 (0.7)	48 (5.9)	239 (22.7)	154 (17.5)	14 (10.1)
Other curative SACT with RT	10 (0.4)	9 (1.1)	37 (3.5)	18 (2.0)	6 (4.3)
Other SACT with RT	0 (0.0)	<5	7 (0.7)	7 (0.8)	0 (0.0)
SACT alone	41 (1.8)	56 (6.8)	185 (17.6)	277 (31.5)	42 (30.2)
No treatment received	114 (5.0)	87 (10.6)	182 (17.3)	181 (20.6)	49 (35.3)
Otherwise unclassified	<5	<5	9 (0.9)	0 (0.0)	0 (0.0)

Abbreviations: RT: radiotherapy; SACT: systemic anticancer therapy.

## Data Availability

The data from this study are not publicly available and no individual data sharing is allowed.

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
