# Peer review of "Real-World Treatment Patterns and Survival Outcomes for Patients with Non-Metastatic Non-Small-Cell Lung Cancer in Sweden: A Nationwide Registry Analysis from the I-O Optimise Initiative"

_cancers, 2024, doi:10.3390/cancers16091655_

Round 1
Reviewer 1 Report
Comments and Suggestions for Authors
The authors have presented treatment patterns and survival outcomes among patients with non-metastatic NSCLC using data from the Swedish National Lung Cancer Registry to establish a baseline before the advent of targeted immunotherapy treatments are introduced in regular practice. The authors presented detailed results by Stage and among all patients. I appreciate the authors discussion of the results. I have the following minor comments:
1. Line 92: IO was not expanded when mentioned for the first time here. Is this the same as I-O for immuno-oncologic? If yes, the please use the same abbreviation.
2. Above comment will address the same abbreviation in Line 105.
3. Lines 114-128: Was this a part of journal guidelines, which was retained for some reason. The authors should remove this section.
4. Same as above for Section 6: Patents
5. For Figures 3-6 the treatment subgroup should be specified with the years (2014-2019) as the figures should be interpreted individually without reviewing the text for details about what the treatment subgroup was.
6. Lines 397-401: I appreciate the author's inclusion of this explanation for patients with Stage IIIC.
7. Line 444: Did the authors mean how the data are reported from routine clinical practice to the registry and that there was no process of adjudication to confirm diagnoses? If yes, ignore comment, if not, please include the above to explain that it may have lead to incomplete or misclassified data.
Author Response
|
Reviewer 1 – Comments |
Response |
|
The authors have presented treatment patterns and survival outcomes among patients with non-metastatic NSCLC using data from the Swedish National Lung Cancer Registry to establish a baseline before the advent of targeted immunotherapy treatments are introduced in regular practice. The authors presented detailed results by Stage and among all patients. I appreciate the authors discussion of the results. I have the following minor comments:
|
We would like to thank the reviewer for their comments and suggested clarifications. |
|
1. Line 92: IO was not expanded when mentioned for the first time here. Is this the same as I-O for immuno-oncologic? If yes, the please use the same abbreviation.
|
The text has been edited so that ‘IO’ is now defined on line 85 as shown: …immunotherapy (IO)-based and…. |
|
2. Above comment will address the same abbreviation in Line 105.
|
I-O in line 105 is used in the context of a Bristol Myers Squibb–led initiative (I-O Optimise) and, as such, is not an abbreviation; it is instead hyphenated to differentiate from the use of IO, which is used as an abbreviation for immunotherapy in the manuscript.
|
|
3. Lines 114-128: Was this a part of journal guidelines, which was retained for some reason. The authors should remove this section |
We would like to apologize for the oversight and can confirm that the boiler plate text has been removed.
|
|
4. Same as above for Section 6: Patents
|
We would like to apologize for the oversight and can confirm that the boiler plate text has been removed.
|
|
5. For Figures 3-6 the treatment subgroup should be specified with the years (2014-2019) as the figures should be interpreted individually without reviewing the text for details about what the treatment subgroup was.
|
Figures 3–6 and appendix figures A1 and A2 have been updated with the following addition to each figure title: (patients diagnosed between January 1, 2014 and December 31, 2019). |
|
6. Lines 397-401: I appreciate the author's inclusion of this explanation for patients with Stage IIIC.
|
We would like to thank the reviewer for their comment. |
|
7. Line 444: Did the authors mean how the data are reported from routine clinical practice to the registry and that there was no process of adjudication to confirm diagnoses? If yes, ignore comment, if not, please include the above to explain that it may have lead to incomplete or misclassified data.
|
We would like to thank the reviewer for their query and can confirm that the intention was to describe how data are reported from routine clinical practice to the registry and that there was no process of adjudication to confirm diagnoses.
|
Reviewer 2 Report
Comments and Suggestions for Authors
This paper presents a pivotal study, utilizing national registry data, to shed light on the treatment and survival patterns of lung cancer patients.The paper is meticulously structured, detailing the study's use of comprehensive national registry data.
Here are some specific points that the authors could elucidate or modify for a more impactful paper:
- There are some parts of text in the manuscript that do not belong to the article but to the journal layout guidelines and should be removed: see lines 114-128 and 468-469.
- Throughout the text, there are many acronyms; perhaps a summary list would be helpful to retrieve them quickly.
- The definition of SACT should be mentioned at the beginning. I suggest moving the definition from line 163 to line 161.
- I wonder why, with national data available, statistical comparisons between different treatments and survival were not considered. The work would have gained more interest.
- It seems that Figure 1 and Figure 4 are related. Would it be interesting to put them side by side in a single illustration?
- How much time elapses from diagnosis to treatment? Are the subjects who received treatment included in Figure 2?
Author Response
Reviewer 2 – Comments |
Response |
|
This paper presents a pivotal study, utilizing national registry data, to shed light on the treatment and survival patterns of lung cancer patients. The paper is meticulously structured, detailing the study's use of comprehensive national registry data. Here are some specific points that the authors could elucidate or modify for a more impactful paper:
|
We would like to thank the reviewer for their comments and suggested clarifications. |
|
There are some parts of text in the manuscript that do not belong to the article but to the journal layout guidelines and should be removed: see lines 114-128 and 468-469.
|
We would like to apologize for the oversight and can confirm that the boiler plate text has been removed.
|
|
Throughout the text, there are many acronyms; perhaps a summary list would be helpful to retrieve them quickly.
|
The abbreviations have been defined on first use throughout the text, according to the journal guidelines.
|
|
The definition of SACT should be mentioned at the beginning. I suggest moving the definition from line 163 to line 161.
|
The abbreviation for SACT has been moved as suggested. |
|
I wonder why, with national data available, statistical comparisons between different treatments and survival were not considered. The work would have gained more interest.
|
Survival analyses were statistically analysed by treatment type using Kaplan-Meier methods and have been presented in Figure 4 (the overall treatment group) and by stage (Supplementary Figures A1a–e) and described in Results section 3.3 (lines 234–266).
|
|
It seems that Figure 1 and Figure 4 are related. Would it be interesting to put them side by side in a single illustration?
|
Figure 1 provides a graphical representation of an analysis of initial treatment patterns using descriptive statistics and, as such, has been shown separately from the Kaplan-Meier survival analysis performed by treatment type, which is shown in Figure 4. As the statistical methodologies used are different, and the results are presented in separate sections of the manuscript, we have chosen to present these figures separately to avoid any potential confusion for the reader.
|
|
How much time elapses from diagnosis to treatment? Are the subjects who received treatment included in Figure 2?
|
Figure 2 represents all patients who were recorded with a diagnosis of NSCLC in the National Lung Cancer Registry (NLCR) between January 1, 2008 and December 31, 2019. Treatment data were only available from 2014; therefore, although patients receiving treatment are included in the overall population, a detailed analysis of treatment received is only possible for patients diagnosed from January 1, 2014 onward. Further explanation of how the data were captured is provided in lines 120–127.
|
Reviewer 3 Report
Comments and Suggestions for Authors
The authors delve into the data available in the Swedish lung cancer national registry focusing on treatment options and outcomes for non metastatic NSCLC. The theme is important given the prevalence, natural evolution and economic impact of the disease.
Here are some suggestions on how to improve the manuscript:
Lines 114 127 and 467 469 contain boiler plate text – should be removed
Figure 4 and associated results are debatable and potential misleading – as survival is most likely linked not only to therapeutic options but also stage, age, comorbidities.
The authors discuss the issue of TNM changing overtime – it would be useful to present the differences.
Given the fact that adenocarcinoma seems the most prevalent variant it would be interesting to see if there were changes on OS and other measures following the advent of EGFR and alk targeted therapies
Looking at the baseline characteristics: most cases are stage I (and stage II) – which is a remarkable performance – is this the result of screening or other type of active detection? Some data on the diagnostic/screening protocols would be particularly interesting.
Age and ECOG status should have been probably taken into account when assessing therapeutic options vs staging – at least when surgery or concomitant chemo/radio is considered.
Conclusions seem underdeveloped and not tailored to the results.
Author Response
Reviewer 3 – Comments |
Response |
|
The authors delve into the data available in the Swedish lung cancer national registry focusing on treatment options and outcomes for non metastatic NSCLC. The theme is important given the prevalence, natural evolution and economic impact of the disease. Here are some suggestions on how to improve the manuscript: |
We would like to thank the reviewer for their comments and suggested clarifications. |
|
Lines 114 127 and 467 469 contain boiler plate text – should be removed |
We would like to apologize for the oversight and can confirm that the boiler plate text has been removed.
|
|
Figure 4 and associated results are debatable and potential misleading – as survival is most likely linked not only to therapeutic options but also stage, age, comorbidities. |
The data reported in Figure 4 and the associated results are based on Kaplan-Meier analyses of survival by treatment type with the aim of providing an overview of patient outcomes by treatment type prior to the availability of targeted therapies and immunotherapies. The analysis performed here did not therefore consider other factors such as comorbidities; wording to address this has been included in the Discussion in lines 429–432, as follows: The data available from this source provide detailed coverage of the baseline characteristics of patients for all hospitals in Sweden; however, the coverage of treatment data with available follow-up information is around 62% of the patient population, and the available data do not include details of patient ethnicity, body mass index, or socioeconomic status.
|
|
The authors discuss the issue of TNM changing overtime – it would be useful to present the differences. |
Unfortunately, we do not have the data available to provide a temporal analysis of changes caused by reclassification of TNM stages. The wording provided between lines 437 and 441 addresses this issue as a limitation of the study, as shown below: The changes in TNM staging that occurred during the course of this study limited the ability to perform a direct comparison of patient outcomes by stage; some analyses were also restricted by sample sizes due to the introduction of new categories (e.g., stage IIIC was introduced in the 8th TNM staging edition), making it challenging to interpret OS data over a longer period of time.
|
|
Given the fact that adenocarcinoma seems the most prevalent variant it would be interesting to see if there were changes on OS and other measures following the advent of EGFR and alk targeted therapies |
The data from the NLCR analyzed here predate the availability of targeted therapies. The date range for this study was selected to provide a baseline upon which future analyses can assess the impact of targeted therapies and immunotherapies on patient outcomes.
|
|
Looking at the baseline characteristics: most cases are stage I (and stage II) – which is a remarkable performance – is this the result of screening or other type of active detection? Some data on the diagnostic/screening protocols would be particularly interesting. |
We would like to thank the reviewer for their comment. In Sweden, as in most other countries, the majority of patients are diagnosed with stage IV disease. As patients with stage IV disease were not included in this study, the most frequently observed stages were stages I and II. In the Discussion section of the manuscript, we compare the distribution observed in our study with results from other studies (lines 334–337).
The wording in the Discussion is as follows: The distribution of patients across initial treatment type by disease stage reported here is similar to that reported in a recent analysis of a large dataset in the US (based on Surveillance, Epidemiology, and End Results and National Program of Cancer Registries data) during the same period [20].
We can confirm that there are no screening programs for lung cancer in Sweden.
|
|
Age and ECOG status should have been probably taken into account when assessing therapeutic options vs staging – at least when surgery or concomitant chemo/radio is considered. |
Thank you for this comment, as this is a retrospective study of data collected in the NLCR, we unfortunately do not have information on the rationale behind treatment decisions made by clinicians at the time. The results therefore represent a historical view of treatment patterns and the associated patient outcomes over a defined period of time.
|
|
Conclusions seem underdeveloped and not tailored to the results. |
As the journal guidelines stipulate that conclusions should be no more than 1–2 paragraphs, we have concluded with an overview of the results presented in this registry study rather than duplicate sections already covered in the Discussion section.
|
Reviewer 4 Report
Comments and Suggestions for Authors
The authors performed statistical analyses on 17433 non-small cell lung carcinoma cases. Cancer registry shows that 53% of lung cancers are diagnosed when the cancer already metastasized and 5-year survival is 8.2%. Treatment options depend mostly on the stage of disease. Most of the lung cancer cases are in stage III meaning that they are unresectable or they do not receive any treatment. The authors showed also overall survival for all stages and time to next treatment or death. The analysis shows how fatal this malignancy is. The study is very interesting because it shows that the most frequent malignancy in men is partly incurable. The study was done on men and women prior to the introduction the newer therapy including immune checkpoint inhibitors and tyrosine kinase inhibitors which should improve the outcome of treatment. The study shows also how bad is lung cancer without good screening and effective treatment. The No of patients in the study is big enough to validate the statistical analyses.
Author Response
Reviewer 4 – Comments |
Response |
|
The authors performed statistical analyses on 17433 non-small cell lung carcinoma cases. Cancer registry shows that 53% of lung cancers are diagnosed when the cancer already metastasized and 5-year survival is 8.2%. Treatment options depend mostly on the stage of disease. Most of the lung cancer cases are in stage III meaning that they are unresectable or they do not receive any treatment. The authors showed also overall survival for all stages and time to next treatment or death. The analysis shows how fatal this malignancy is. The study is very interesting because it shows that the most frequent malignancy in men is partly incurable. The study was done on men and women prior to the introduction the newer therapy including immune checkpoint inhibitors and tyrosine kinase inhibitors which should improve the outcome of treatment. The study shows also how bad is lung cancer without good screening and effective treatment. The No of patients in the study is big enough to validate the statistical analyses. |
We would like to thank the reviewer for their comments. |
Reviewer 5 Report
Comments and Suggestions for Authors
The criteria for inclusion and exclusion of patients should be better clarified separately .
Were intermediate histotypes such as adenosquamous considered?
Since the stage I is the prevalent stage I suggest to include the types of surgery , the complications and propaedeutic evaluations. It would be interesting to establish the predictors of 5-year survival in the early stages
I suggest to discuss about functional and clinical evaluations of patients undergoing surgery .
I suggest to include the following manuscript for the discussion
J Int Med Res. 2022 Jun;50(6):3000605221094531.
Author Response
|
Reviewer 5 – Comments |
Response |
|
|
We would like to thank the reviewer for their comments and suggested clarifications. |
|
The criteria for inclusion and exclusion of patients should be better clarified separately.
|
The following additional wording has been added to lines 119–120 to clarify the exclusion criteria for the study: Patients with missing data on age at diagnosis or sex were excluded.
|
|
Were intermediate histotypes such as adenosquamous considered?
|
These data were not available in the registry for inclusion in the study. |
|
Since the stage I is the prevalent stage I suggest to include the types of surgery, the complications and propaedeutic evaluations. It would be interesting to establish the predictors of 5-year survival in the early stages.
|
These data were not available in the registry for inclusion in the study. |
|
I suggest to discuss about functional and clinical evaluations of patients undergoing surgery.
|
We would like to thank the reviewer for this comment; however, as this is a retrospective study of data collected in the NLCR we unfortunately do not have detailed information on functional and clinical evaluations of patients made by clinicians at the time. The results therefore represent a historical view of treatment patterns and the associated patient outcomes over a defined period of time.
|
|
I suggest to include the following manuscript for the discussion: J Int Med Res. 2022 Jun;50(6):3000605221094531.
|
We would like to thank the reviewer for the manuscript recommendation. As the focus of the Pezzuto et al. publication is a targeted study of the variables associated with post-surgery respiratory failure, the content appears to be more specific than the results presented in our paper. We unfortunately do not have the appropriate available data in the NLCR to provide a meaningful discussion of these results in this context and have therefore not included the publication in the discussion.
|